# Long-Term Outcome of Patients with TPO Mutations

**DOI:** 10.3390/jcm10173898

**Published:** 2021-08-30

**Authors:** Leraz Tobias, Ghadir Elias-Assad, Morad Khayat, Osnat Admoni, Shlomo Almashanu, Yardena Tenenbaum-Rakover

**Affiliations:** 1Pediatric Department B, Ha’Emek Medical Center, Afula 1834111, Israel; 2Pediatric Endocrine Institute, Ha’Emek Medical Center, Afula 1834111, Israel; ghadir_ea@clalit.org.il (G.E.-A.); admoni_o@clalit.org.il (O.A.); rakover_y@clalit.org.il (Y.T.-R.); 3Technion Institute of Technology, Rappaport Faculty of Medicine, Haifa 3200003, Israel; 4Genetic Institute, Ha’Emek Medical Center, Afula 1834111, Israel; Morad_kh@clalit.org.il; 5The National Newborn Screening Program, Ministry of Health, Tel-Hashomer, Ramat Gan 5262100, Israel; Sholomo.Almashanu@sheba.health.gov.il

**Keywords:** congenital hypothyroidism, dyshormonogenesis, goiter, thyroid peroxidase (TPO) enzyme, multinodular goiter, *TPO* mutation

## Abstract

Introduction: Thyroid peroxidase (TPO) deficiency is the most common enzymatic defect causing congenital hypothyroidism (CH). We aimed to characterize the long-term outcome of patients with TPO deficiency. Methods: Clinical and genetic data were collected retrospectively. Results: Thirty-three patients with primary CH caused by TPO deficiency were enrolled. The follow-up period was up to 43 years. Over time, 20 patients (61%) developed MNG. Eight patients (24%) underwent thyroidectomy: one of them had minimal invasive follicular thyroid carcinoma. No association was found between elevated lifetime TSH levels and the development of goiter over the years. Conclusions: This cohort represents the largest long-term follow up of patients with TPO deficiency. Our results indicate that elevated TSH alone cannot explain the high rate of goiter occurrence in patients with TPO deficiency, suggesting additional factors in goiter development. The high rate of MNG development and the risk for thyroid carcinoma indicate a need for long-term follow up with annual ultrasound scans.

## 1. Introduction

The incidence of congenital hypothyroidism (CH) is estimated at 1:2000–3000 births [1]. In 2020, the European Society for Pediatric Endocrinology updated guidelines for diagnosis and management for CH [2]. Thyroid dysgenesis accounts for 75–80% of the cases, and thyroid hormone synthesis disorders (dyshormonogenesis) account for 10–15% of them [3]. Thyroid dyshormonogenesis is commonly inherited in an autosomal recessive pattern and is more prevalent in highly consanguineous populations. Thyroid peroxidase (TPO) deficiency is the most common enzymatic defect, accounting for about 50% of the cases [4]. TPO catalyzes both the iodination and coupling of tyrosyl residues of thyroglobulin with a strict requirement for hydrogen peroxide, which acts as an electron acceptor [5]. A mutation in *TPO* was first reported in 1990 [6] (OMIM # 274500); since then, over 60 different mutations have been identified, mainly missense mutations (http://portal.biobaseinternational.com/hgmd/pro/, accessed on 18 June 2009) [5].

*TPO* mutations are present in primary CH with a normal-size or goitrous thyroid gland. The development of a goiter occurs either in utero, at birth, or during life. It has been postulated that the development of a goiter over the years can be attributed to elevated thyroid-stimulating hormone (TSH), resulting in hyperplasia and hypertrophy of the thyrocytes. TSH acts through stimulation of the TSH receptor in two pathways, one regulating thyroid growth, and the other regulating transcription of the genes encoding Na^+^/I^−^ symporter, thyroglobulin, TPO, and pendrin to produce thyroid hormones. Elevated serum TSH levels are associated with increased risk of thyroid cancer [7]. In the past, TPO enzyme defects were diagnosed mainly by perchlorate discharge test. However, molecular sequencing of the *TPO* gene has rendered this test irrelevant. Several studies have described the molecular findings of patients with *TPO* mutations [1,4,8,9,10,11,12], but only a few them have reported the long-term clinical outcome [13]. Here, we report retrospectively on the long-term follow up (up to 43 years) of 33 patients with *TPO* mutations and evaluate the effect of TSH serum levels on goiter development.

## 2. Patients and Methods

### 2.1. Patients

From a cohort of 278 subjects with CH being followed at our institute, 41 (15%) were diagnosed with dyshormonogenesis. Among them, 33 patients (80%) aged 4 to 43 years had TPO enzyme deficiency and were enrolled in the study. The diagnosis of TPO deficiency was made clinically by the findings of primary CH, elevated serum thyroglobulin levels, and a normally placed thyroid gland demonstrated by ^99^TC scan or by ultrasound (US) imaging. I131-uptake and a perchlorate discharge test were performed in some of the patients prior to the availability of molecular sequencing. Final diagnosis was made by sequencing the *TPO* gene. In the case of a known mutation in the family, sequencing of the specific mutation was performed. In five subjects, the diagnosis was based on clinical characteristics or the presence of at least one additional affected member in the family. Patients were followed up every 6 months and levothyroxine (LT4) therapy doses were adjusted to maintain TSH and free T4 (FT4) levels within the normal ranges. When receiving abnormal results, L-T4 dose was adjusted and repeat tests were performed after 6 weeks from L-T4 dose change. The presence of a goiter was determined by physical examination of the thyroid gland by the physician and was confirmed by US imaging. Data were collected retrospectively from these patients’ medical files and included: clinical, biochemical, imaging, and molecular findings from birth up until the time of the study.

### 2.2. Biochemical Analyses

Thyroid hormone levels at birth were obtained from the national neonatal screening records. Laboratory thyroid function at birth, at annual follow up, and at the time of the study were collected retrospectively from the medical files. Thyroid hormones levels were obtained at 0800 am prior to administering LT_4_ therapy. To assess the effect of elevated TSH on goiter development, we used three different parameters: (i) mean FT_4_ serum levels, (ii) median lifetime TSH serum levels, and (iii) percentage of lifetime that TSH and FT_4_ were in the normal range (normal TSH, referred to as TSH below 5 mIU/L; normal FT4, referred to as FT4 between 10–20 pmol/L).

Blood samples were collected by heel puncture 48–72 h after birth. The Israeli CH screening program is based on total T_4_ (TT_4_) level followed by confirmatory TSH test, such that when the TT_4_ level is below the 10th percentile for age, TSH is measured. TSH values >20 mIU/L are considered indicative of primary CH.

### 2.3. Hormone Analyses

TSH and FT_4_ were measured by direct automated chemiluminescent immunoradiometric assay using the ADVIA Centaur immunoassay system (Bayer Corporation, Tarrytown, NY, USA). TSH reference values provided by our laboratory were 0.4–4.2 mIU/L and FT_4_ normal reference was 10–20 pmol/L.

### 2.4. Imaging

The size and location of the thyroid gland was demonstrated by ^99^TC scan at the age of 2–3 years, after withdrawing from LT_4_ therapy for 3 weeks. In some patients, a repeat ^99^TC scan was performed due to the development of MNG or the presence of a thyroid nodule. Patients underwent US imaging yearly to detect goiter development. Computerized scans (CT) were performed in cases when tracheal pressure was suspected by US scan or prior to surgery.

### 2.5. Molecular Analysis

After signing the appropriate informed consent, peripheral blood samples were collected for DNA analysis. Genomic DNA was extracted from peripheral mononuclear cells using the Blood Amp Kit (QIAGEN Inc., Valencia, CA, USA), according to the manufacturers’ protocol. Sequencing of the *TPO gene* was performed. In the case of a known mutation in the family, sequencing of the specific mutation identified in the family was performed.

### 2.6. Statistical Analysis

Statistical analyses were performed with SAS v9.4 statistical software package (Cary, NC, USA). Categorical variables were presented as prevalence and percent, and continuous variables as mean, median, standard deviation (SD), and ranges. Study groups were compared by Kruskal–Wallis test (continuous variables) and Chi-square or Fisher’s Exact test (categorical variables). Risk factors for goiter development were analyzed by univariate and multivariate logistic regression analyses (age at time of the study, mean FT4 and median TSH, and percentage of lifetime abnormal results of TSH and FT4). Nonparametric regression (Loess) was used to analyze the association between serum TSH levels and goiter development. Significance value was set at *p* < 0.05.

## 3. Results

Thirty-three patients were enrolled in the study. Their characteristics are presented in Table 1. All were of Arab-Muslim origin, with a high rate of consanguinity, belonging to 7 extended families (15 core families). All subjects were living in the same region in the northeast of Israel in seven neighboring villages. CH was diagnosed by national neonatal screening in 25 patients (75%), of which 7 (23%) were diagnosed clinically prior to obtaining newborn screening results. Two patients were born before the implementation of national newborn screening in Israel, and screening results were not available for 6 patients (Table 1).

At diagnosis, the presenting symptoms were prolonged jaundice, macroglossia, and umbilical hernia (Table 2). At 1 year of age, 10 children (30%) had delayed psychomotor development (Table 2). At the time of the study, 9 patients (27%) had neurological sequelae; among them, 7 patients had mental retardation and 4 patients had sensorineural hearing impairment.

Congenital goiter was reported in 4 patients (Table 2), but the goiter in these cases resolved after initiation of LT_4_. Over the years, 20 patients developed MNG (Table 2). Most of the patients had a huge goiter as shown in Figure 1, and all had at least twice the size of a normal gland. Among them, 8 patients (40%) underwent thyroidectomy due to either local tracheal pressure or histological findings by fine needle aspiration suspecting thyroid carcinoma.

### 3.1. Imaging Results

All patients had a normally placed thyroid gland as shown by ^99^TC scan, and in 3 patients (12.5%), a hypertrophic gland was demonstrated at the age of 2–3 years. A second scan was performed in 10 patients and 60% showed an enlarged MNG with either cold or warm nodules. Eight patients (24%) underwent total thyroidectomy (Table 3). All patients had benign follicular adenoma or hypertrophy apart from case 14, who was diagnosed with follicular carcinoma (Table 3).

### 3.2. Genetic Results

Three different mutations were identified, with either homozygous or compound heterozygous inheritance. Nineteen patients were homozygous for c.1618C>T p.Arg540Ter, 4 patients were homozygous for c.1477G>A, p.Gly493Ser, and 4 patients were compound heterozygous for both mutations. Two siblings were homozygous for the c.875C>T p.Ser292Phe mutation. When examining the presence of goiter in each mutation, we have found that 13 patients (68%) were homozygous for the c.1618C>T p.Arg540Ter mutation, 4 patients (100%) were homozygous for c.1477G>A, p.Gly493Ser, and 2 patients (100%) were homozygous for the c.875C>T p.Ser292Phe mutation. No compound heterozygous patients developed a goiter. No association was shown between the type of mutation and goiter development.

### 3.3. Effect of TSH Values on Development of MNG

Comparing patients with and without a goiter for mean FT4 and median TSH levels revealed no differences between the two groups (Table 4). Patients without a goiter had a higher percentage lifetime TSH level (TSH above 5 mIU/L) than those with a goiter. TSH levels were high in both groups, implicating low compliance to therapy, though in most patients, FT_4_ levels were in normal range during the study (normal reference was 10–20 pmol/L.) However, patients with a goiter were significantly older than those without a goiter (Table 4). No associations were found between the TSH parameters and the need for thyroidectomy.

## 4. Discussion

We report on the clinical outcome of 33 patients with TPO deficiency for an extensive period of up to 43 years. We show that 61% of our patients developed MNG over time. The development of MNG was not associated with higher TSH values, indicating that elevated TSH levels have only a minor impact on goiter development. Among the 8 patients that underwent thyroidectomy, 1 patient had minimally invasive follicular carcinoma and the other 7 had either follicular adenoma or hyperplasia.

Among patients diagnosed with dyshormonogenesis, 33 (80%) had TPO deficiency. The incidence of TPO deficiency in our group was higher than that described previously. In Japan, the incidence is as low as 13%, Portugal 23% [11,12], and in Slovakia, Serbia, and the Czech Republic, 45% [4]. The high occurrence of *TPO* mutations in our cohort was attributed to the high rate of consanguinity in our population (80%).

The clinical presentation of *TPO* mutations at referral was similar to that of CH from other etiologies. Most patients (80%) presented with signs of hypothyroidism in the first days of life prior to obtaining the neonatal screening report. The main signs were prolonged jaundice, macroglossia, umbilical hernia, and coarse facial features indicating that hypothyroidism may have already existed in utero.

Congenital goiter was observed in 4 infants (12%). This finding is in contrast to a French cohort in which congenital goiter was reported in 60% of patients [14]. The high rate of congenital goiter in the French study might be attributed to the fact that the ^99^TC scan was performed immediately after birth, whereas in our study, the presence of goiter relied on physical examination, while the ^99^TC scan was performed at 2–3 years of age.

Three different mutations in the *TPO* gene were identified. The c.1618C>T p.Arg540Ter mutation was reported previously in populations from Japan and the Netherlands [12,15], c.1477G>A p.Gly493Ser was reported in patients from China and Portugal [9,11], and the c.875C>T p.Ser292Phe mutation was reported in patients from Tunis [10]. In family members without *TPO* mutations, there was no predisposition to developing a goiter.

Most of our patients (61%) developed a goiter over time, at an average age of 8.6 years. Goiter was determined by physical examination and followed by US imaging yearly once diagnosed [16]. Thyroid antibodies including anti-TPO, and anti-thyroglobulin was negative. Eight patients underwent thyroidectomy. In 5 of them, local pressure on the trachea, proven by CT scan, was the indication for surgery, whereas the other 3 patients had histological findings in fine needle aspiration of suspected malignancy. The pathological findings revealed histological characteristics of dyshormonogenesis [17]. An exception was patient 14, who was diagnosed with minimally invasive follicular carcinoma. This case was previously reported by our group [18] and it joins few other reports of follicular carcinoma in patients with *TPO* mutations [19,20]. Mice implanted with a thyroid gland that continuously produced TSH showed a high percentage of malignant transformations [21].

A high TSH level has been postulated to promote thyroid hypertrophy and hyperplasia, and increases the risk for thyroid malignancy [18,22]. We evaluated the association of MNG development and lifetime TSH levels in our cohort. The lifetime TSH levels were above the normal range in the entire cohort, indicating nonadherence to LT_4_ therapy in both groups. However, a comparison of patients with and without a goiter did not reveal higher lifelong TSH levels in patients with MNG. These results contradict the common assumption that goiter development is mainly dependent on serum TSH levels. Moreover, no association was found between a specific mutation and the development of goiter. We assume that goiter development in patients with *TPO* mutations is a result of a combination of genetic and environmental factors. Iodine is the main environmental factor interacting with TPO, and therefore, it has been speculated that iodine levels could modify the phenotype of patients affected with *TPO* mutations [5]. With regard to iodine deficiency as a parameter that may induced goiter development in our cohort, there is sufficient iodine in the diet in the Northern region of Israel. Interestingly, a previous study has shown increased risk of differentiated thyroid carcinoma in European populations with certain *TPO* gene variants [23].

Seven patients had a diagnosis of mental retardation, 3 of them severe. We assume that the severe neurological sequelae in these patients can be attributed mainly to a delay in diagnosis and initiation of therapy, and not to the etiology of *TPO* mutation. Four patients had sensorineural hearing impairment. A high rate of hearing impairment in CH has been shown recently by our group [24]. Furthermore, *TPO* mutations have been reported to be associated with sensorineural deafness in humans [25] as well as in animal models [26], indicating that the TPO enzyme itself may have a role in the development of the auditory system.

Finally, we should note this study’s limitations: (i) older patients in our cohort had delayed initiation of supplemental therapy, and (ii) guidelines for CH management have changed over the years [2,27].

In summary, we show a high rate of MNG in patients with *TPO* mutations, some of whom required thyroidectomy. No association was found between goiter development and elevated TSH, indicating that in addition to high TSH values, goiter development is attributed to other genetic and environmental factors. One patient developed follicular thyroid carcinoma, whereas all the others had hyperplastic adenomatous goiter consistent with dyshormonogenesis.

## 5. Conclusions

This study describes the long-term clinical outcome of a large cohort of patients with *TPO* mutations, emphasizing the development of MNG over the years. The high rate of MNG development and the risk for thyroid carcinoma indicates the need for long-term follow up and annual US scans in these patients.

## Figures and Tables

**Figure 1 jcm-10-03898-f001:**
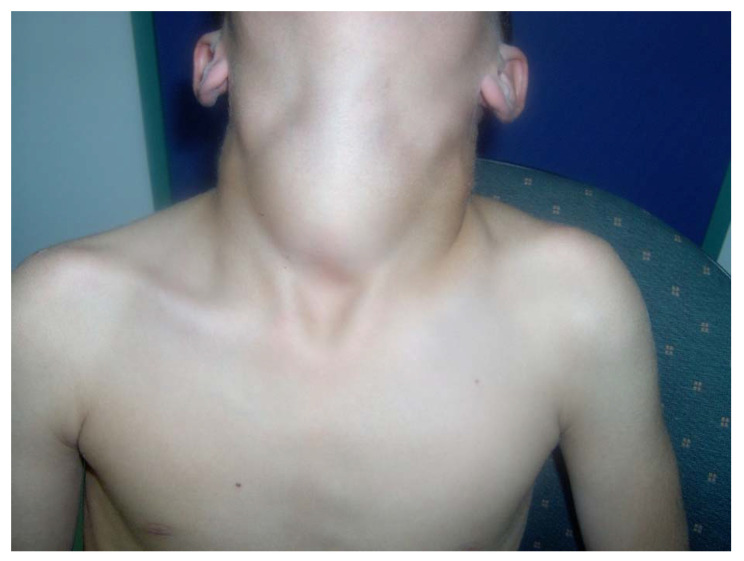
Massive MNG in a 15-year-old male with *TPO* mutation. (Written informed consent was obtained from the patient for publication of this case report).

**Table 1 jcm-10-03898-t001:** Patient characteristics.

Patient no./ Family no.	Gender	Age at Diagnosis (Days)	Neonatal Screening ^1^		Age at Study (Years)	Consanguinity
			TSH (mIU/L)	TT_4_ (µg/dL)		
1/1	M	5	500	0.2	11	Y
2/1	F	2	258	1.3	8	Y
3/2	M	23	200	0.7	29	N
4/2	F	4	250	2	17	Y
5/2	M	3	200	1.8	22	Y
6/3	M	14	NA	NA	36	Y
7/3	F	47	183	1.5	27	Y
8/4	F	28	200	2.11	35	N
9/4	F	24	NA	NA	27	Y
10/4	M	20	NA	NA	13	N
11/4	M	7	200	2	28	N
12/5	M	20	200	2.3	22	Y
13/6	F	7	308	2.7	5	Y
14/6	F	7	301	1.9	9.7	Y
15/7	M	6	350	2	10	Y
16/7	M	5	557	2.3	4	Y
17/8	M	5	135	6.9	26	Y
18/8	F	13	228	1	33	Y
19/8	F	7	200	1.9	21	Y
20/9	F	28	NA	NA	37	Y
21/9	F	28	200	1	34	Y
22/9	F	10	200	1.3	29	Y
23/10	M	17	200	1	28	N
24/11	M	98	NA	NA	41	Y
25/11	M	17	200	1.1	22	Y
26/12	F	7	500	2.4	9	N
27/12	F	7	452	2.4	6	N
28/13	F	270	ND	ND	43	Y
29/14	F	60	ND	ND	38	Y
30/15	M	17	200	0.9	27	Y
31/14	M	1	200	2.5	23	Y
32/14	M	6	128	3	12	Y
33/15	M	3	NA	NA	8	Y
Mean ± SD (Range)	17 M/16F	24.2 ± 51 (1–252)	273 ± 126 (127–558)	1.9 ± 1.3 (0.2–6.9)	21.7 ± 12.3 (4–43)	26 Y (78.8%)

^1^ Screening TSH normal range <20 mIU/L; screening TT_4_ normal range 10th percentile for age. M, male; F, female; Y, yes; N, no; NA, not available; ND, not done.

**Table 2 jcm-10-03898-t002:** Clinical characteristics of all patients with TPO enzyme deficiency.

Symptoms and Signs at Presentation	No of Patients (%)
Prolonged jaundice	12 (36)
Macroglossia	9 (27)
Umbilical hernia	9 (27)
Coarse facial features	8 (24)
Hypotonicity	6 (18)
Large fontanel	4 (12)
Goiter	4 (12)
Hypothermia	2 (6)
None	4 (12)
Neurological outcome at 1 year	
Normal psychomotor development	20 (60)
Delayed motor skills	4 (12)
Delayed language skills	2 (6)
Delayed combined motor & language skills NA	4 (12) 2 (6)
Goiter	
MNG	20 (60)
Age at which goiter first observed (years)	8.76 ± 6.06 (0.9–22)

NA, not available; MNG, multinodular goiter.

**Table 3 jcm-10-03898-t003:** Summary of imaging and pathology of patients that underwent thyroidectomy.

Patient no.	Age (Years) ^1^	Imaging	FNA	Pathology	Post-Surgery Complication
5	34	Two enlarged lobes with non-homogeneous contour. Calcifications in right lobe. Both lobes have pressure effect on airway (CT)	ND	Multinodular adenomatoid gland	None
12	15	Enlarged nodule in right lobe, liquid-filled with multiple septations (US)	Follicular cystic lesion with cellular atypia (Bethesda 3)	Follicular adenoma	None
13	16	MNG—both lobes enlarged with heterogeneous contour. Enlarged lymph nodes (US)	Multiple follicular cells with enlarged nucleus—suspected follicular tumor (Bethesda 4)	Follicular hyperplasia nodular goiter	Hypoparathyroidism Nephrolithiasis
14	25	MNG with pressure on trachea (CT)	ND	Minimally invasive follicular carcinoma of left lobe	None
15	15	Diffuse enlargement of the thyroid gland with hypodense cystic areas enlarged into upper mediastinum to 1 cm below the sternal notch (CT)	Follicular hyperplasia (Bethesda 2)	Adenomatoid hyperplasia Goiter	Hypoparathyroidism
20	19	MNG (US)	ND	Adenomatoid gland	None
27	15	Large MNG with pressure on trachea (CT)	Follicular cells with some metaplastic changes (Bethesda 3)	NA	Hypoparathyroidism
31	12	Large MNG with cold nodules with calcifications (US)	Lymphoid hyperplasia (Bethesda 2)	Hyperplastic nodular gland	None

^1^ At time of study. FNA, fine needle aspiration; NA, not available; ND, not done; CT, computerized scan; US, ultrasound scan; MNG, multinodular goiter.

**Table 4 jcm-10-03898-t004:** Comparison of thyroid function parameters between patients that developed goiter and patients without goiter.

	Goiter		*p*-Value
	No	Yes	
	N = 13	N = 20	
Age at study (years)	14.21 ± 11.38 (4–38)	27.8 ± 7.99 (11–43)	0.0022
Lifetime TSH (mIU/L) ^1^	34% ± 20% (11–67)	51% ± 17% (29–82)	0.0212
Lifetime FT_4_ (pmol/L) ^2^	98% ± 4% (85–100)	90% ± 17% (33–100)	0.1073
Median TSH (mIU/L)	12.54 ± 12.16 (2.3–43.35)	6.75 ± 4.54 (1.48–18.05)	0.2311
Mean FT_4_ (pmol/L)	20.94 ± 6.62 (17–42.08)	17.88 ± 4 (5.99–24.27)	0.2343

Mean ± SD (Range); ^1^ Percentage of life that TSH was in the normal range (referred to as TSH below 5 mIU/L). ^2^ Percentage of life that FT_4_ levels were in the normal range (referred to as FT_4_ between 10–20 pmol/L).

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
