# Peer review of "Long-Term Outcome of Patients with TPO Mutations"

_jcm, 2021, doi:10.3390/jcm10173898_

Round 1

Reviewer 1 Report

SUMMARY:

This study reports the clinical findings of 33 patients with congenital hypothyroidism secondary to TPO mutations. In particular, authors correlate the trends of their thyroid function over the years with the development of goitre, and conclude that thyroid function does not influence the development of goitre. This conclusion contrasts with what has been commonly observed in the literature and might reflect a not good enough study design where some possible confounding factors were not taken into account, as detailed below.

STRENGHTS:

  • Clearly written
  • Thyroid analysis including both biochemical and imaging tools
  • Longitudinal design

WEAKNESSES:

  • Small number of patients
  • Retrospective study
  • Lack of important information: i.e. details about levothyroxine treatment, number and frequency of thyroid tests, other considered risk factors for goitre (i.e. iodine intake and thyroid autoimmunity), ecc …

SPECIFIC COMMENTS

  • In the material and methods authors should clarify that this is a retrospective study; otherwise they communicate the wrong message that this is a prospective study up to 43 years of follow-up (i.e. lines 55-56).
  • Line 115: authors mention that they took into consideration the risk factors for goitre development, however they don’t list them. Furthermore, they mention that multivariate logistic regression analyses have been performed (lines 115-116), however in Tab 4 only univariate analysis has been shown.
  • Table 4. I don’t understand the meaning of percentages of lifetime TSH and FT4 within the normal range. For example does 0.34 means that patients had normal values of TSH only during the 0.34% of their life, thus they always had abnormal thyroid function? Why were they not started on treatment with levothyroxine? Or why the dosage of levothyroxine was not adjusted to normalise TSH/FT4 values? Furthermore, in the manuscript details about the levothyroxine treatments are missing.
  • FT4 usually has a Gaussian distribution, while TSH has a skewed distribution, thus authors should use only mean FT4 and median TSH, removing median FT4 and mean TSH from the analysis and Tab 4.
  • Authors should report how lifetime FT4 and TSH have been calculated. How many blood tests had every patient during his/her life? How frequently? Were they always performed in the same lab and with the same assay? I doubt this is possible, especially for patients with longer follow-up.
  • Since this is a study analysing the goitre, authors should mention in this patient cohort the urinary iodine concentrations, iodine intake and presence or absence of anti-thyroid peroxidase and anti-thyroglobulin antibodies.
  • Tab 4: in the multivariate analysis authors should also correct for other risk factors for goitre (i.e. urinary iodine and iodine intake, presence of thyroid autoimmunity, etc), the age at diagnosis of CH (and thus the start of LT4 treatment), the age of thyroid assessment and the adherence/compliance to LT4 treatment. However with such small number of patients, the inclusion of such variants (all important) in the analysis will weaken the statistics.
  • Definition of goitre: what are the volume’s cut off considered, adjusted for age? Please report them.
  • Furthermore, it would be interesting to report the numeric value of thyroid gland’s volume. Even if TSH/FT4 are not related with dichotomy presence/absence of goitre, the concentrations of TSH/FT4 might correlate with thyroid’s volume.
  • In the discussion authors conclude that the type of TPO mutation does not correlate with the presence/absence of goitre, however I cannot see such analysis in the results.
  • Considering the above study limitations, authors should be more cautious in concluding for an absence of TSH/FT4 effect on the goitre development, until a more complete analysis will be performed.

Reviewer 2 Report

Leraz et al. perfomed a retrospective analysis of 33 patients with congenital hypothyroidism (CH) due to the TPO gene mutations living in the area of the northeast Israel. The authors analyzed in detail biochemical, imaging and clinical features of the disease over a long term follow-up period. The study is well-designed and it provides original data, that may be helpful for the clinical management of CH patients with TPO mutations. The reviewer highly assess the manuscript and has no major comments.

Author Response

Thank you for your review.

Round 2

Reviewer 1 Report

Authors have still not addressed some of my comments, as detailed below:

1)OK

2) Line 121: please modify “mean and median TSH and FT4” to “mean FT4 and median TSH”.

3) Ok but I did not mean the definition of “percentages”, I meant their presentation. I assume that that for example 0.34 refers to 34%? Reporting data as % would be easier to follow for readers.

4) I meant details about the levothyroxine treatment, for example dose and therapy duration. But since you did not report them, I assume you don’t have these data.

5) Median FT4 and mean TSH are still present in some parts of the manuscript, for example at lines 86-87 specify only median TSH and mean FT4, removing mean TSH and median FT4. Please double check the rest of the manuscript.

6) Ok

7) Ok

8) Ok

9) Authors did not answer my question. What cut-offs for thyroid volumes did you use, according to the various age groups?

10) I fully agree that the volume of thyroid gland changes in different age groups. This is why defining a precise cut-off volume for each age group is important to define the presence or absence of goitre, otherwise this assessment risks to become arbitrary, especially if different radiologists performed the ultrasound scans and especially in only mildly enlarged thyroid glands, that without a precise cut-off can be arbitrarily considered both “normal” or goitre, depending on the operator. The presence or absence of goitre is a key finding in this study and thus should be defined as more precisely as possible, especially considering that authors reach conclusions that contrast with what has been observed before. Please provide additional details. The lack of the precise thyroid volume is a major limitation; if you don’t have saved images to re-calculate it (preferable option), at least describe an indicative size of goitre (twice as normal, three times as normal, etc …) in your population.

11) Authors did not provide the requested results. How many patients with each mutation did, or did not, present a goitre? Authors should provide details for each of the three mutations, in the text or in a table. For example, of the 19 patients homozygous for c.1618C>T p.Arg540Ter, N (%) had goitre and N (%) had not. The same should be done for the other mutations.

12) Ok
